

# Observations of nitrated phenols in four sites in North China: Concentrations, source apportionment, and secondary formation

Liwei Wang[1], Xinfeng Wang[1], Rongrong Gu[1], Hao Wang[1], Lan Yao[1], Liang Wen[1], Fanping Zhu[2], Weihao Wang[3], Likun Xue[1], Lingxiao Yang[1,2], Keding Lu[4], Jianmin Chen[1,2,5], Tao Wang[3], Yuanghang Zhang[4], and Wenxing Wang[1]

[1]Environment Research Institute, Shandong University, Ji'nan, Shandong 250100, China
[2]School of Environmental Science and Engineering, Shandong University, Ji'nan, Shandong 250100, China
[3]Department of Civil and Environmental Engineering, Hong Kong Polytechnic University, Hong Kong, China
[4]State Key Laboratory for Environment Simulation and Pollution Control, College of Environmental Sciences and Engineering, Peking University, Beijing 100871, China
[5]Shanghai Key Laboratory of Atmospheric Particle Pollution and Prevention, Department of Environmental Science and Engineering, Institute of Atmospheric Sciences, Fudan University, Shanghai 200433, China

*Correspondence to*: Xinfeng Wang (xinfengwang@sdu.edu.cn)

**Abstract.** Samples of fine particulate matters were collected at four sites in North China (urban, rural, and mountain) in summer and winter, and the contents of nine nitrated phenols were quantified in the laboratory with the use of ultra-high-performance liquid chromatography coupled with mass spectrometry. During the sampling periods, the concentrations of particulate nitrated phenols exhibited distinct temporal and spatial variation. On average, the total concentrations of particulate nitrated phenols in urban Ji'nan in the wintertime reached 48.4 ng m$^{-3}$, and those in the summertime were 9.8, 5.7, 5.9, and 2.5 ng m$^{-3}$ in urban Ji'nan, rural Yucheng and Wangdu, and Mt. Tai, respectively. The elevated concentrations of nitrated phenols in wintertime and in urban areas demonstrate the apparent influences from anthropogenic sources. The positive matrix factorization receptor model was then applied to determine the origins of particulate nitrated phenols in North China. The five major source factors were traffic, coal combustion, biomass burning, secondary formation, and aged coal combustion plume. Among them, coal combustion played a vital role, especially at the urban site in the wintertime, with a contribution of around 55%. In the summertime, the observed nitrated phenols were highly influenced by aged coal combustion plumes at all of the sampling sites. Meanwhile, in remote areas,



contributions from secondary formations were significant. Further correlation analysis indicates that nitrosalicylic acids were produced mostly from secondary formations that were dominated by $NO_2$ nitration.

## 1 Introduction

Nitrated phenols, with at least one nitro and one hydroxyl group attached on the benzene ring, have effects on the environment and on living beings that cannot be ignored. Due to their strong absorption in the near-ultraviolet and visible regions, nitrated phenols are classified as poorly characterized "Brown Carbon" (BrC) (Desyaterik et al., 2013; Teich et al., 2017). Though the absorption of BrC is weak compared to that of Black Carbon (BC), it can enhance the absorption of solar radiation and may have an indirect effect on regional climate (Feng et al., 2013; Mohr et al., 2013; Laskin et al., 2015; Lu et al., 2015; Zhao et al., 2015). Recent study has shown that several semivolatile nitrated phenols, including nitrophenol, nitrocatechol, and methylnitrocatechol, are important photochemical oxidation products of gas phase precursors, which may play a part in the formation of secondary organic aerosols (SOA) (Pereira et al., 2015). Via photo-oxidation and aqueous phase reactions, nitrated phenols could alter the tropospheric photochemistry and influence the secondary organic aerosol formation and thus affect regional air quality and atmospheric visibility (Mohr et al., 2013; Laskin et al., 2015; Chow et al., 2016; Yuan et al., 2016). In addition, nitrated phenols are toxic and hazardous to plants, animals, aquatic life, and human beings (Natangelo et al., 1999; Schüssler and Nitschke, 2001; Harrison et al., 2005a; Ganranoo et al., 2010). The significant roles of nitrated phenols in the atmosphere and their adverse effects on living things have aroused the urgent need to understand their pollution characteristics and their origins.

Nitrated phenols in the troposphere largely originate from primary emissions from anthropogenic activities. They have drawn attention in recent decades owing to their prevalence in biomass burning plumes (Kahnt et al., 2013; Mohr et al., 2013), which have significant effects on regional air quality and climate. Due to the good correlation with levoglucosan, Iinuma et al. (2010) pointed out that methylnitrocatechols could be a reasonable tracer for biomass burning events. In addition to biomass burning, other combustion activities such as coal combustion and motor vehicle operation are possible



sources of nitrated phenols (Nojima et al., 1983; Tremp et al., 1993; Yan et al., 2015); their contributions, however, have not yet been well evaluated. Furthermore, nitrated phenols have been used as pesticides in some farmlands and widely used as raw materials in industries.

Secondary formations also play important roles in the atmospheric abundance of nitrated phenols (Grosjean, 1985). Various atmospheric processes are responsible for the formation of nitrated phenols, including photochemical oxidation of benzenes in the presence of $NO_x$ (NO and $NO_2$) and ultraviolet rays (Nojima et al., 1975, 1976), oxidation of phenols by OH and $NO_3$ radicals in the presence of $NO_x$ (Atkinson et al., 1992; Bolzacchini, et al., 2001), and nitration of phenols in aqueous solutions (Lüttke et al., 1997; Vione et al., 2005; Frka et al., 2016). In particular, the characteristics and importance of the formation pathways regarding the condensed phase remain more uncertain than those for the gas phase (Harrison et al., 2005a). Generally, nitration of phenols, which takes place in both gas and condensed phases, could be the most important process identified. Precursors such as phenols and cresols are emitted and produced via photochemical oxidation of benzenes, and nitration processes then occur with the participation of $NO_2$. During this process, the phenols and nitrated phenols partition from time to time between the gas and particle phases according to their saturated vapor pressure (Nojima et al., 1975; Bolzacchini et al., 2001; Harrison et al., 2005b; Xu and Wang, 2013). Chow et al. (2016) found evidence from field measurements that high levels of nitrated phenols in Hong Kong were associated with high concentrations of $NO_2$.

North China, an important economic and industrial center of East Asia, experiences severe $PM_{2.5}$ pollution and high levels of $NO_x$ (Wang et al., 2014; Wang et al., 2015a; Zhang et al., 2015). In recent years, several field studies related to concentrations, size distributions, and light absorption of particulate nitrated phenols have been conducted in and near this region, showing the large influences from anthropogenic sources such as biomass burning and other combustion activities (Li et al., 2016; Teich et al. 2017). In this study, we present the concentrations, compositions, and temporal and spatial differences of nine nitrated phenols in samples of $PM_{2.5}$ collected from four different sites in North China. Source apportionment was applied to understand the origins of the particulate nitrated phenols, and five major sources were identified. The results highlight the large contribution of coal combustion





at the urban site, especially during winter, and the important role of secondary formation in remote areas during summer.

## 2 Experiments and methods

### 2.1 Sampling sites and on-line instruments

$PM_{2.5}$ filter sampling and related online measurements were conducted in five field campaigns at four sampling sites in North China, which are illustrated in detail in Table 1. As shown, two field campaigns were conducted in urban Ji'nan in winter 2013 and summer 2014, whereas the measurements at the rural and mountain sites took place in summer 2014.

The sampling site in urban Ji'nan was the Atmospheric Environment Observation Station of

Shandong University (AEOS-SDU) (N 36°40', E 117°03'). The urban Ji'nan site is surrounded by education, commercial, and residential districts, with several roads with heavy traffic nearby. Therefore, it is frequently influenced by vehicle emissions, urban plumes, and industrial plumes from suburban areas. The $NO_x$ concentration was measured by a chemiluminescence method equipped with a molybdenum oxide convertor (Model 42C, TEC, USA). The $O_3$ concentration was analyzed with an

ultraviolet absorption method (Model 49C, TEC). The SO2 concentration was measured by the pulsed fluorescence method (Model 43C, TEC). Water-soluble ions, including $Cl^-$, $NO_3^-$, $SO_4^{2-}$, $NH_4^+$, $K^+$, and $Ca^{2+}$, were detected with the online Monitor for AeRosols and GAses in ambient air (MARGA, ADI20801, Applikon-ECN, Netherlands). The online instruments and methods used for the winter field campaign were described in detail by Wang et al. (2015b). In the summer campaign, due to logistical

issues, water-soluble ions were determined from quartz-filtered samples of $PM_{2.5}$ by ion chromatography (Wang et al., 2017a).

The second sampling site is located in an open field in rural Yucheng (N 36°52', E 116°34'), Dezhou, Shandong Province, about 50 km northwest from the city of Ji'nan (see Fig. 1). During summer, polluted urban plumes arrive with the assistance of prevailing winds from the southeast. The rural

Yucheng site is surrounded by farmland, but there is a national road 1.5 km away to the south and a few small-scale factories nearby. Trace gases of $NO_x$ and $O_3$ were measured with Model 42C and Model



49C analyzers, respectively. Water-soluble ions were detected with the MARGA. Details about the site and the instruments deployed were described by Yao et al. (2016).

The third sampling site is located in rural Wangdu (N 38 ˚42', E 115 ˚08'), Baoding, Hebei Province, near grasslands and gardens. Nevertheless, it can be easily influenced by industrial and urban plumes from large cities like Beijing, Tianjin, and Shijiazhuang. Trace gases of $NO_x$ and $O_3$ were measured with Model 42i analyzer coupled with a blue light convertor and Model 49i analyzer (TEC). Water-soluble ions were analyzed by a gas aerosol collector–ion chromatography (GAC-IC) system. More information on the site and instruments may be found in Tham et al. (2016).

The fourth sampling site is located almost at the summit of Mt. Tai (N 36°15', E 117°06'), Tai'an, Shandong Province. It is less frequently influenced by incense burning and restaurants due to a distance of ~1km from the famous tourism spots at Mt. Tai. Notably, with an altitude of about 1465 m above sea level, the Mt. Tai site is an ideal place to understand the transport, sources, and formation processes of air pollution in North China. Trace gases of $NO_x$ and $O_3$ were measured with a Model 42i analyzer coupled with a blue light convertor and a Model 49i analyzer, respectively. Water-soluble ions were analyzed with the MARGA. Detailed information of this site and the deployed instruments was given by Wang et al., (2017b).

## 2.2 $PM_{2.5}$ sampling

$PM_{2.5}$ filter samples were collected using the mid-volume sampler (TH-150A, Wuhan Tianhong, China) with a flow rate of 100 LPM (liters per minute). The sampler was fixed in the open fields with the $PM_{2.5}$ cutting head 1.5 m above the ground or station. Two filter samples were collected each day, with the daytime samples from 07:00 to 18:30 (local time) and the nocturnal samples from 19:00 to 06:30 the next day. The filter samples were restored under −20°C until subsequent mass weighing and chemical analysis of nitrated phenols, organic carbons, and water-soluble ions. The collected fine particulate matter was weighed with a microbalance (ME5, Sartorius, Germany) for calculation of the $PM_{2.5}$ concentration. The organic carbon concentration was determined by a thermal-optical method with the protocol of NIOSH 5040 (Sunset, OCEC analyzer, USA) and was further used to estimate the




concentration of organic matter by multiplying by a factor of 1.8 at the urban Ji'nan site and by a factor of 2.0 at the remote sites of Yucheng and Mt. Tai (Aiken et al., 2008; Yao et al., 2016).

**2.3 Analytical method of nitrated phenols**

The preparation procedure and analytical method of nitrated phenols used in this study were adopted

from those developed by Kitanovski et al. (2012) and Chow et al. (2016). First, the $PM_{2.5}$ filter sample was extracted with 20 ml methanol (containing a small amount of EDTA) twice for 10 min. The extracts were evaporated with a rotary evaporator to near dryness. After they were re-dissolved to about 2 ml, the extracts were filtered through a polytetrafluoroethylene syringe filter (0.45 μm; 3 mm). The sample solution was then dried with a gentle stream of nitrogen and re-dissolved to 300 μL with

methanol containing 200 μg·L$^{-1}$ 2,4,6-trinitrophenol as an internal standard.

The determination of nitrated phenols was performed with an Ultimate 3000 ultra-high-performance liquid chromatography system coupled with an LCQ Fleet mass spectrometer (UHPLC-MS, Thermo Scientific, USA) and equipped with an electrospray ionization source. The nitrated phenols were separated with an Atlantis T3 C18 column (2.1 × 150 mm, 3-μm particle size, Waters, USA) with a

15 guard column Atlantis T3 (2.1 × 10 mm, 3-μm particle size). The mobile phase contained 11% acetonitrile in methanol (A) and 11% acetonitrile and 0.1% acetic acid in water (B). The proportion of elution A started at 34% and gradually increased to 66% within 19 min. It was then kept at 66% for 4 min and then decreased to 34% for the last 8 min.

The mass signals at six mass-to-charge ratios (138, 152, 154, 168, 182, and 228 amu) were monitored

under the selective ion mode, and nine kinds of nitrated phenols were identified: 4-nitrophenol (4NP), 2-methyl-4-nitrophenol (2M4NP), 3-methyl-4-nitrophenol (3M4NP), 4-nitrocatechol (4NC), 4-methyl-5-nitrocatechol (4M5NC), 3-methyl-5-nitrocatechol (3M5NC), 3-methyl-6-nitrocatechol (3M6NC), 3-nitro-salicylic acid (3NSA), and 5-nitro-salicylic acid (5NSA). Standards of these compounds and other isomers are applied for chemical identification and further quantification.



# 3 Results and discussion

## 3.1 Temporal and spatial variations

During the sampling periods, the concentrations of nitrated phenols in PM$_{2.5}$ presented distinct seasonal and diurnal variations. As shown in Table 2, in urban Ji'nan, the concentrations of total nitrated phenols (ΣNPs) in the wintertime were much higher than those in the summertime. The mean concentration in the wintertime in urban Ji'nan reached 48.4 ng m$^{-3}$, about five times the mean concentration at this site in the summertime (9.8 ng m$^{-3}$). Meanwhile, the proportion of ΣNPs in PM$_{2.5}$ and the ratio of ΣNPs to organic matter were both high in the wintertime (average, 0.28‰ and 1.69‰, respectively), more than three times those in the summertime. In addition, the diurnal pattern of nitrated phenols differed in winter and summer. During the winter campaign, both the concentration and the proportion of nitrated phenols in PM$_{2.5}$ at night largely outweighed those during the day. In the summertime, however, no significant difference was observed. The above differences in the concentration and proportion of particulate nitrated phenols with seasonal and diurnal changes were possibly associated with the emission sources and formation pathways and with meteorological factors such as solar radiation, temperature, and relative humidity.

The abundance of nitrated phenols also exhibited large variations among locations and site types. In the summertime, the concentrations of nitrated phenols in urban Ji'nan, rural Yucheng, rural Wangdu, and Mt. Tai were 9.8±4.2, 5.7±2.8, 5.9±3.8, and 2.5±1.6 ng m$^{-3}$, respectively (also shown in Table 2). The higher concentration of ΣNPs in the urban site than those measured at the rural and mountain sites indicates significant contributions from anthropogenic activities. The concentrations of particulate nitrated phenols observed in North China in this study were comparable with those measured in North and East China in previous studies and substantially higher than those found in Germany (Li et al., 2016; Teich et al., 2017).

Spatial differences were also observed in the proportions of the various kinds of nitrated phenols (shown in Fig. 2). Among them, nitrocatechols and methylnitrocatechols (NCs), which are most abundant in biomass burning plumes (Wang et al. 2017c), appeared in lower concentrations than the other nitrated phenols in this study. Nitrophenols and methylnitrophenols (NPs), however, were predominant in the measurements at the Ji'nan (40%) and Wangdu sites (56%). The domination of NPs



instead of NCs in particulate nitrated phenols at the urban site indicates that a large fraction of the nitrated phenols originated from sources other than biomass burning. The strong correlations between NPs and NCs (see Table 3) suggest that they might have similar sources. At the rural and mountain sites, the proportions of nitrosalicylic acids (NSAs) were much higher than that in urban Ji'nan. In particular, in Yucheng and Mt. Tai, NSAs contributed about half of the concentration of ΣNPs. The NSAs, which contain carboxyl groups, are naturally more oxidized than the other nitrated phenols measured in this study. They exhibited lower correlation coefficients (lower than 0.6) than most other nitrated phenols (also shown in Table 3). It is interesting that 5NSA does not correlate well with 3NSA (correlation coefficient, 0.57), indicating that these two components might have different origins or experience different formation processes.

Overall, obvious temporal and spatial variations were observed in the concentrations and compositions of nitrated phenols in North China. The implied differences in the primary sources and the secondary formation pathways are analyzed in detail in the following sections.

## 3.2 Source apportionment

### 3.2.1 Source identification

To obtain a comprehensive understanding of the sources of nitrated phenols observed in North China, the positive matrix factorization (PMF) receptor model was applied with the measurement data from the nine nitrated phenols and from nine additional tracer species. Ninety-one sets of input data were used and the model was run 40 times to choose the optimal solution. Based on the results of the PMF model, five major factors were finally identified and are shown in Fig. 3.

The first source factor, traffic, had the highest concentration of NO and contributed an average of 17% of the total nitrated phenols. This source mainly includes emissions from heavy traffic in the urban areas of Ji'nan and motor vehicles on the roads near the Yucheng and Wangdu sites. The direct emission of nitrated phenols from automobile motors was confirmed in a laboratory study by Tremp et al. (1993), who gave an emission factor of 9 to 36 nmol/L.

The second source factor, which is featured with the highest loading of $SO_2$ and particulate sulfate, was identified as coal combustion, which accounted for 15% of the nitrated phenol concentrations and





included direct emissions from coal combustion from industries and residential usage. In previous field measurements at the summit of Great Dun Fell, UK, coal combustion was considered to be associated with the relatively high levels of nitrated phenols in cloud (Lüttke et al., 1997).

The third source factor, biomass burning, was identified with highest concentrations of potassium and chloride and high level of particulate nitrate in $PM_{2.5}$. This source contributed 18% to the total particulate nitrated phenols. It mainly included biomass burning in nearby farmlands and rural areas and cooking smoke from restaurants and residential kitchens.

The fourth source factor, secondary formation, showed high concentrations of $O_3$ (one of the major oxidants in the troposphere) and fine particulate nitrate, sulfate, and ammonium. It contributed 23% to the concentrations of nitrated phenols. Secondary nitrated phenols were produced from precursors along with the formation of $O_3$ and secondary inorganic aerosols.

The last source factor, aged coal combustion plume, showed relatively high levels of both $SO_2$ and $O_3$ and accounted for 27% of the nitrated phenol concentration. This source, which was also related to coal combustion, was ascribed to the aging processes of coal combustion products.

### 3.2.2 Contributions of primary sources at different sites and in different seasons

To clarify the difference in the sources of particulate nitrated phenols at the four sampling sites in North China, the average contributions of the five source factors to the concentrations of total particulate nitrated phenols in summertime and wintertime are compared in Figure 4.

Biomass burning, which in previous studies was usually regarded as the most important source of nitrated phenols, was identified in all five field measurements in this study. However, it did not present as the dominant contributor at any site, which was attributed to the very strict control measures on open biomass burning activities, comprehensive utilization of biomass, and improved biomass burning techniques in household stoves in recent years (Lin et al., 2015). As shown in Fig. 4, the average contributions of biomass burning in the summertime followed the order Jinan (21%) > Yucheng (18%) ≈ Wangdu (18%) > Mt. Tai (10%), which are comparable with the contribution of 17% in the wintertime in Ji'nan.



The influence of traffic on the concentrations of particulate nitrated phenols was observed during all sampling periods and played an important role in the summertime. In particular, traffic source contributed the most (27%) in urban Ji'nan in the summertime, possibly because of the heavy traffic load nearby. The contributions of traffic in rural Yucheng and Wangdu were 12% and 20%, respectively, mainly due to the national and high-speed roads nearby, which could be busy at nighttime with more diesel vehicles. The nitrated phenol levels at Mt. Tai were seldom affected by traffic transportation, with an average contribution of only 3%.

Surprisingly, coal combustion emission and its aged plumes, which received little attention in previous studies, were identified in all five field measurements and often dominated the sources of nitrated phenols. In the wintertime in urban Ji'nan, about 55% of the particulate nitrated phenols originated from coal combustion. This contribution was approximately three times that in the summertime in Ji'nan. Therefore, the huge contribution of coal combustion could be the main cause of the elevated nitrated phenol concentrations in the wintertime and the high demand of residential heating at nighttime might lead to the elevated nocturnal concentrations and proportions of nitrated phenols in $PM_{2.5}$ in cold season. In addition, the contribution of aged coal combustion to nitrated phenols was more important in the summertime than in wintertime and reached 37% in Wangdu and 40% at Mt. Tai. The large contribution of aged coal combustion at rural sites in the summertime was likely linked to the rapid oxidation processes of gas-phase products of coal combustion from nearby industries with high atmospheric oxidation capacity in the summertime in North China.

As coal still serves as the dominant energy resource in North China and the demand from residential heating increases dramatically during the winter, it is possible that the massive consumption of coal resulted in elevated levels of nitrated phenols in this region. Although a series of regulations have been published and strict emission reduction measures have been implemented in the past decade, coal combustion remains a major source of air pollution in North China. Therefore, pertinent strategies and control techniques against coal combustion are still needed to reduce the concentrations of nitrated phenols as well as their precursors and consequently to mitigate the effects on the climate and the environment.



### 3.3 Secondary formation of nitrated phenols

As mentioned above, secondary formation was recognized as an important source of particulate nitrated phenols in North China, especially in the summertime. It was identified as the dominant contributor in the field measurements in rural Yucheng (41%) and Mt. Tai (42%). To clarify the secondary formation of nitrated phenols and to understand the underlying chemical processes, the relationships between nitrated phenols and the related pollutants (including nitrate, $NO_2$, and $O_3$) were analyzed in detail by discarding the samples with high contributions (>40%) from primary emission sources (including biomass burning, traffic, and coal combustion) based on the results of PMF model.

Fine inorganic nitrates, which are mainly produced via homogeneous and heterogeneous oxidations of $NO_x$, are among the most important secondary formed nitrogen-containing constitutes (Wen et al., 2015). Scatterplots of the concentrations of fine nitrates and nitrated phenols are presented in Fig. 5. It can be seen that the concentrations of nitrated phenols increased with the increasing nitrate contents at the three remote sites. The good correlations and trends between them indicate that the nitrated phenols in the selected $PM_{2.5}$ samples in the rural and mountain areas of North China in the summertime originated mainly from secondary formation rather than primary emissions.

To further understand the dominant factors that influence the secondary formation of particulate nitrated phenols, we examined the variations of precursors of $NO_2$ and available oxidants of $O_3$. Figure 6 illustrates the concentrations of nitrated phenols and the averaged mixing ratio of $NO_2$ and $O_3$ at nighttime in the summer campaigns (only data from nocturnal samples were analyzed here with regarding to the photolysis of nitrated phenols in the daytime). In general, higher concentrations of ΣNPs were associated with higher mixing ratio of $NO_2$ at all four sites. The relevance of nitrated phenols to $NO_2$ suggests that $NO_2$ played an important role in the secondary formation of nitrated phenols in North China, and that nitration of phenolic compounds was thus the probable dominant production process. Concurrent variation of nitrated phenols with $O_3$ was also observed in rural Yucheng and at Mt. Tai. Nevertheless, little correlation was found between the concentration of nitrated phenols and the $O_3$ mixing ratio in urban Ji'nan and rural Wangdu, indicating that the oxidant level acted as a limiting factor to the abundance of nitrated phenols only in certain conditions. Further studies are required to obtain a reasonable explanation. As for the daytime samples, however, due to the





influence of photolysis, no apparent relationship was found between nitrated phenols and $NO_2$ or $O_3$ (not shown here).

Fig. 6 shows that distinct differences exist in the concentrations and variations of three kinds of nitrated phenols in the five field measurements. To understand the discrepancy in the formation mechanism among them, we further inspected the correlations of the nocturnal abundances of NPs, NCs, and NSAs with the $NO_2$ concentrations. As shown in Fig. 7, NSAs, which increase with the mixing ratio of $NO_2$, showed a high correlation coefficient at the remote site of Mt. Tai, and they exhibited a moderately good relationship in rural Wangdu. The similar variation trend and strong correlation between NSAs and $NO_2$ at the remote sites and the high proportions of NSAs in the source factor of secondary formation are good evidence that most of the NSAs came from secondary transformation of aromatic precursors. However, only weak correlation was identified between NSAs and $NO_2$ in rural Yucheng (not shown here), where nitration may not be the prevailing aging process in the formation of NSAs. NCs exhibited moderately good correlations with $NO_2$ at the Yucheng and Wangdu sites rather than at the Mt. Tai site, indicating that the secondary produced NCs were also largely associated with anthropogenic emissions and that the formations were probably associated with emissions of precursor volatile organic compounds from combustion of biomasses and fossil fuels. Furthermore, NPs were found to have good correlation with $NO_2$ only at the Mt. Tai site, suggesting that the secondary formation of NPs was controlled by $NO_2$ in conditions of low $NO_x$.

In summary, on the one hand, $NO_2$ played a major role in the formation of nitrated phenols in rural and mountainous areas in North China. On the other hand, because of diverse sources and aging processes, the origins of the three kinds of nitrated phenols were distinguished at different types of sites. Further comprehensive field measurements on nitrated phenols, their precursors, and oxidants are required to understand the formation mechanisms in various atmospheric conditions.

## 4 Summary and conclusions

Nine nitrated phenols in fine particulate matters were measured in four different types of sites in North China. Much higher concentrations of nitrated phenols were observed in the wintertime and in the urban site, indicating strong influences from anthropogenic activities. To clarify the sources of nitrated



phenols in this region, PMF receptor model was applied and five major sources were identified, including traffic, biomass burning, coal combustion, secondary formation, and aged coal combustion plume. Surprisingly, coal combustion contributed significantly to the total concentration of nitrated phenols. It was recognized as the dominant source at the urban site in the wintertime, and aged coal combustion contributed a large fraction at the rural and mountain sites in the summertime. In addition, secondary formation was the most important source of nitrated phenols in the summertime in remote areas and the majority of nitrosalicylic acids might come from secondary formation of the precursor aromatic compounds with the participation of $NO_x$. The similar variation pattern and strong correlation between nitrated phenols and $NO_2$ at the rural and mountain sites suggest that the nitration process was very likely to be the dominant formation pathway.

This work highlights the crucial roles of anthropogenic sources playing in the pollution levels and variation characteristics of nitrated phenols in the atmosphere in North China. Further control measures on emission reduction in both particulate-phase and gas-phase pollutants from coal combustion are needed to mitigate the pollution and the environmental impacts of nitrated phenols in this region. Concurrent measurements of nitrated phenols, the aromatic precursors, and oxidants are required in the future to obtain a comprehensive understanding on the formation mechanisms in various conditions.

### Acknowledgements

This work was supported by National Natural Science Foundation of China (Nos. 41775118, 91544213, 91644214, 21407094) and the National Key Research and Development Program of China (No. 2016YFC0200503). The authors would like to thank Dr. Monique Teich for language corrections and providing valuable suggestions.

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





**Tables:**

**Table 1. Sampling sites and sampling periods involved in this study.**

| Sampling site | Type | Altitude | Sampling time | Season |
|---|---|---|---|---|
| Ji'nan | Urban | ~50 m | 2013.11.26 - 2014.01.05 | Winter |
| | Urban | | 2014.09.04 - 2014.09.21 | Summer |
| Yucheng | Rural | ~23 m | 2014.06.09 - 2014.06.20 | Summer |
| Wangdu | Rural | ~14 m | 2014.06.19 - 2014.06.29 | Summer |
| Mt. Tai | Mountain | ~1465 m | 2014.07.27 - 2014.08.06 | Summer |

**Table 2. Average concentration levels of total particulate nitrated phenols (ΣNPs) and the proportions in PM$_{2.5}$ and fine organic matters.**

| Site | Season | Concentrations (ng m$^{-3}$) | | | NPs/PM$_{2.5}$ (‰) | NPs/OM (‰) |
|---|---|---|---|---|---|---|
| | | Daytime | Nighttime | Average | | |
| Ji'nan | Winter | 38.0±26.3 | 55.8±24.0 | 48.4±25.5 | 0.28±0.09 | 1.69±0.64 |
| Ji'nan | Summer | 10.3±4.9 | 9.3±3.6 | 9.8±4.2 | 0.09±0.04 | 0.41±0.16 |
| Yucheng | Summer | 5.2±2.3 | 6.3±3.2 | 5.7±2.8 | 0.05±0.02 | 0.23±0.09 |
| Wangdu | Summer | 6.0±4.3 | 5.7±3.3 | 5.9±3.8 | 0.07±0.04 | |
| Mt. Tai | Summer | 2.8±1.5 | 2.3±1.8 | 2.5±1.6 | 0.06±0.09 | 0.17±0.14 |

**Table 3. Linear correlation coefficients ($r$) between individual nitrated phenols measured in PM$_{2.5}$ samples in five measurements.**

| | 4NP | 3M4NP | 2M4NP | 4NC | 4M5NC | 3M6NC | 3M5NC | 5NSA | 3NSA |
|---|---|---|---|---|---|---|---|---|---|
| 4NP | 1.00 | | | | | | | | |
| 3M4NP | 0.95 | 1.00 | | | | | | | |
| 2M4NP | 0.95 | 0.96 | 1.00 | | | | | | |
| 4NC | 0.77 | 0.64 | 0.70 | 1.00 | | | | | |
| 4M5NC | 0.86 | 0.81 | 0.82 | 0.82 | 1.00 | | | | |
| 3M6NC | 0.85 | 0.78 | 0.81 | 0.85 | 0.94 | 1.00 | | | |
| 3M5NC | 0.85 | 0.79 | 0.80 | 0.92 | 0.90 | 0.90 | 1.00 | | |
| 5NSA | 0.27 | 0.16 | 0.29 | 0.48 | 0.38 | 0.43 | 0.41 | 1.00 | |
| 3NSA | 0.58 | 0.57 | 0.61 | 0.52 | 0.58 | 0.63 | 0.57 | 0.53 | 1.00 |



**Figure captions:**

**Figure 1.** Maps of the study region show (a) the $NO_x$ emission intensity and (b) the four sampling sites. Squares ■ represent urban site, circles ● represent rural sites, and the triangles ▲ represent mountain site.

**Figure 2.** Fractions of individual nitrated phenols in the total particulate nitrated phenols at four sampling sites in wintertime and summertime. *W* represents the winter measurement and *S* represents the summer measurement.

**Figure 3.** Source profiles of nitrated phenols and related air pollutants obtained from PMF analysis.

**Figure 4.** Contributions of the five source factors to the concentrations of particulate nitrated phenols at

10 four sampling sites in wintertime and summertime. *Total* represents the average contributions of the five field measurements).

**Figure 5.** Scatterplots of particulate nitrated phenols with fine inorganic nitrate.

**Figure 6.** Time series of concentrations of three kinds of nitrated phenols in the nocturnal samples collected in the summertime and the corresponding mixing ratio of $NO_2$ and $O_3$.

**Figure 7.** Scatterplots of nocturnal concentrations of nitrated phenols with $NO_2$. NPs-MT and NSAs-MT indicate samples collected at Mt. Tai, NCs-WD and NSAs-WD samples were collected in Wangdu and NCs-YC samples were collected in Yucheng.



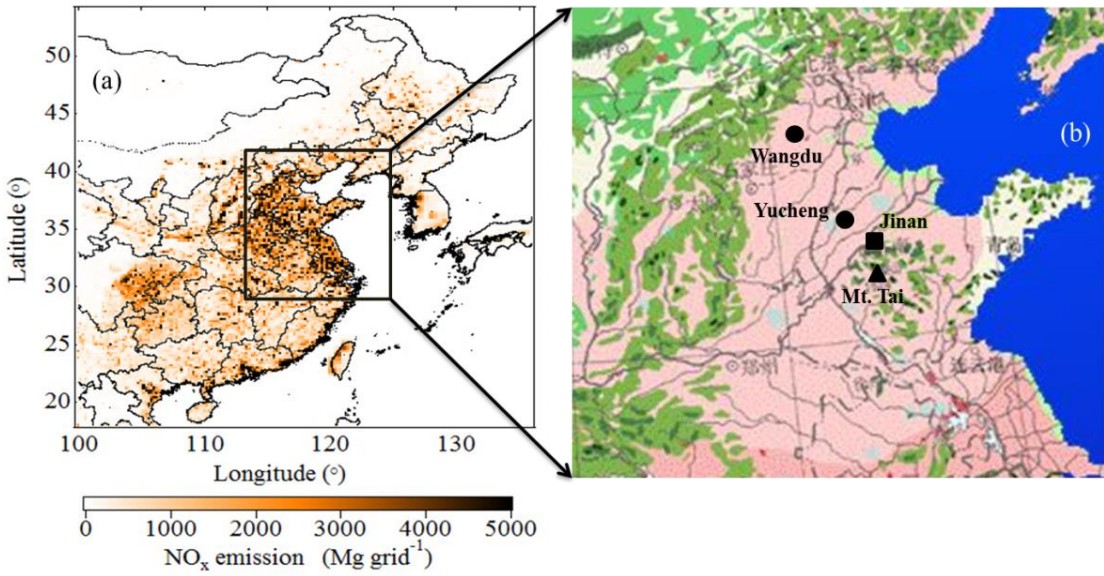

**Figure 1. Maps of the study region show (a) the NO$_x$ emission intensity and (b) the four sampling sites. Squares ■ represent urban site, circles ● represent rural sites, and the triangles ▲ represent mountain site.**

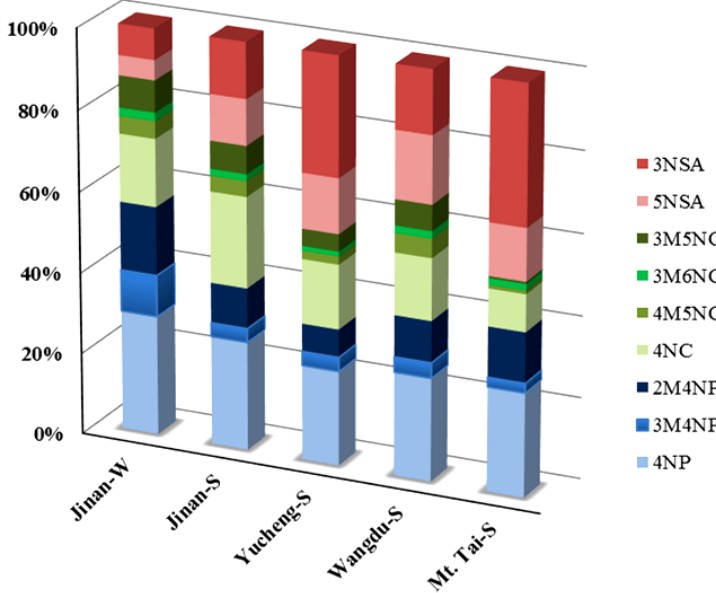

**Figure 2. Fractions of individual nitrated phenols in the total particulate nitrated phenols at four sampling sites in wintertime and summertime. *W* represents the winter measurement and *S* represents the summer measurement.**





**Figure 3. Source profiles of nitrated phenols and related air pollutants obtained from PMF analysis.**



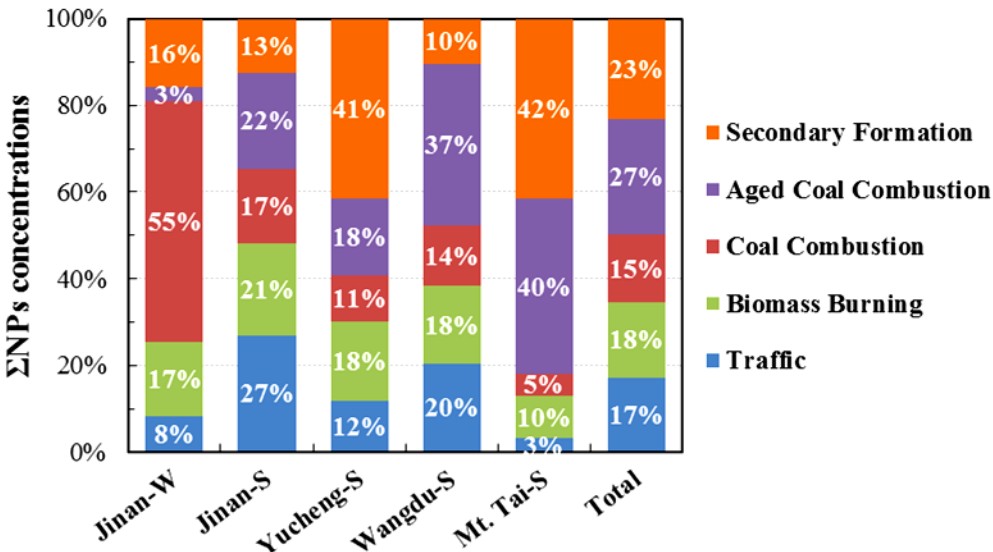

**Figure 4. Contributions of the five source factors to the concentrations of particulate nitrated phenols at four sampling sites in wintertime and summertime.** *Total* **represents the average contributions of the five field measurements).**

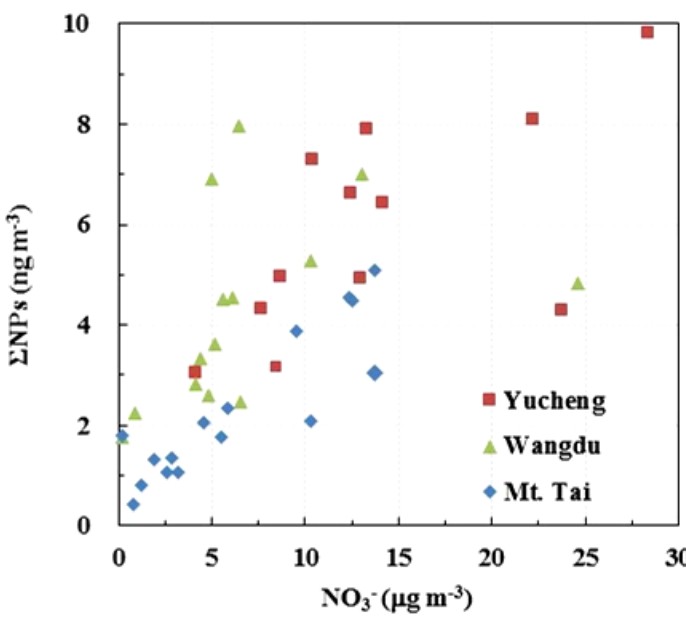

**Figure 5. Scatterplots of particulate nitrated phenols with fine inorganic nitrate.**




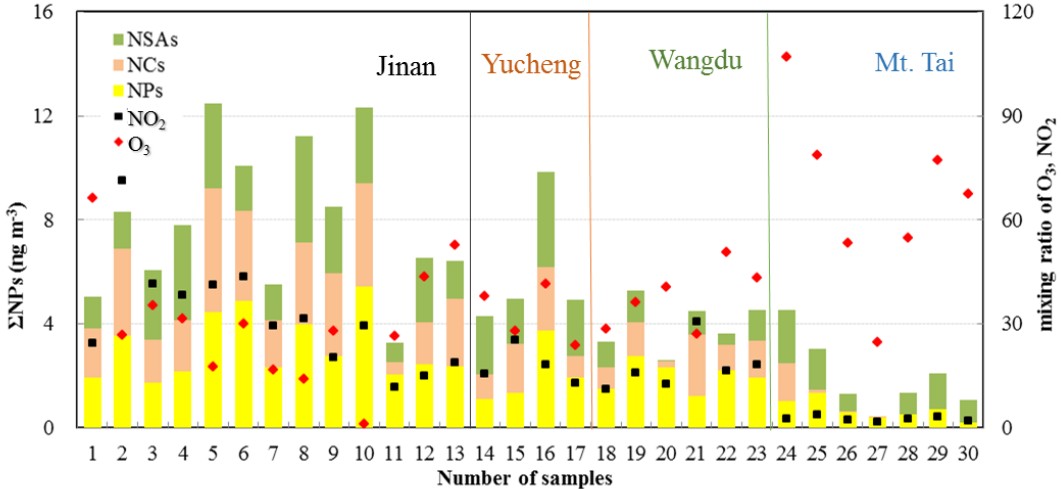

**Figure 6. Time series of concentrations of three kinds of nitrated phenols in the nocturnal samples collected in the summertime and the corresponding mixing ratio of NO$_2$ and O$_3$.**

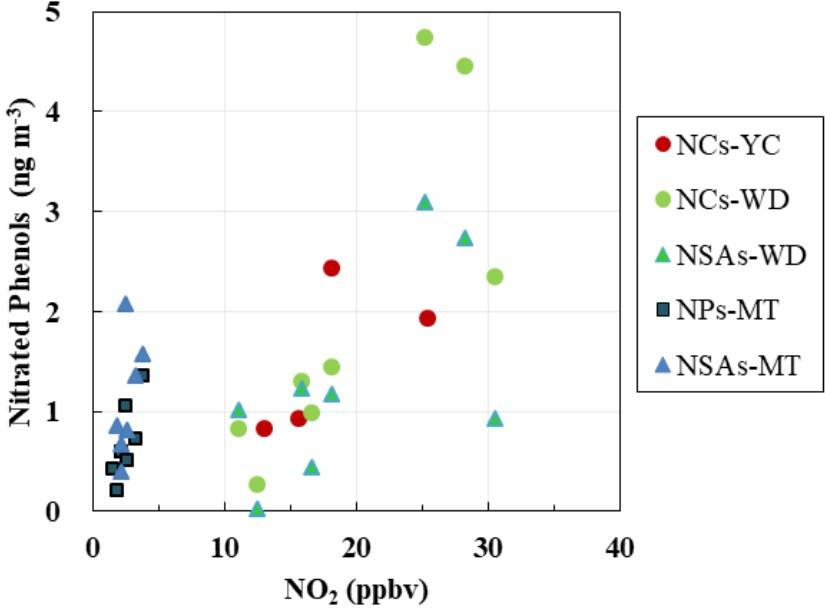

**Figure 7. Scatterplots of nocturnal concentrations of nitrated phenols with NO$_2$. NPs-MT and NSAs-MT indicate samples collected at Mt. Tai, NCs-WD and NSAs-WD samples were collected in Wangdu and NCs-YC samples were collected in Yucheng.**