# Peer review of "Observations of fine particulate nitrated phenols in four sites in North China: Concentrations, source apportionment, and secondary formation"

_Atmospheric Chemistry and Physics, 2017_

## Referee Comment (RC1) · Anonymous Referee #1 · 12 Dec 2017

General: This is an interesting study about measurements of nitrated phenols in northern China. Nitrophenols have been analysed and a source apportionment by PMF has been performed identifiying five main contributing source factors.

The paper is largely correctly written and contains a wealth of valuable information. Nonetheless, the English language of the manuscript should be checked once more, preferably by a native speaker.

Unfortunately, the analytical finding for the NPs are not being linked to aerosol optical propoerties which is a topic currently much discussed but maybe this is foreseen for a follow-up or sister publication.

[Figure]

Overall, I think the paper can be published subject to only minor revision.

Details

Page 5. line 6 :....converter...

Page 8, line 14 ff: Please discuss if there are other PMF solutions which do explain equal fractions but derive another number of factors. For Jinan and Wangdu the sum of the five factors is not 100 %. Please mention and discuss.

Figure captions, Figure 5: Scatter plots (in two words)

Page10, line 8: Please discuss why coal combustion has not been identified before. This may just be due to the fact that this is not a strong source factor contributor either in Shanghai or in Germany.

Page 11, section 3.3.: Maybe this section can be substantiated somewhat by a discussion which chemical mechanisms actually lead to the secondary formation. The section is a bit unspecific. The occurence of which compounds can be explained by pure gas phase processes and where are product observed where multiphase processes could probably be involved ? Is there any correlation with haze occurence ?

---

## Referee Comment (RC2) · Anonymous Referee #2 · 11 Jan 2018

The authors measured several species of nitrated phenols in PM2.5 filter samples at several sites in North China. A positive matrix factorization (PMF) receptor model was applied to investigate the sources of nitrated phenols, which were found to be traffic, coal combustion, biomass burning, secondary formation, and aged coal combustion. Discussion of the secondary sources of various nitrated phenols was included.

I find that this manuscript includes a nice analysis of diurnal, seasonal, and spatial differences in the measured compounds, which are important constituents of organic aerosol. The strength of the manuscript could be greatly improved by including more detail about nitrated phenol species identification, as well as more detail about how

the PMF model was run and how the solutions were chosen. I would recommend this paper for publication after considering the following comments:

General comments:

Title: For clarity, you may want to specify in the title that you measured "particulate nitrated phenols" instead of just "nitrated phenols", as you have done in the abstract and other places.

Pg. 3 Ln. 3: Please add citations to support this sentence.

Pg. 3 Ln. 27-Pg. 4 Ln. 2: These last two sentences in the introduction are actually statements of your results, which should not be included in the introduction. Please consider revising these sentences so that they simply state what you did and are about to present, and not what you found.

Pg. 6 Ln 1-2: Why did you multiply by 1.8 and 2.0?

Pg. 6 Ln. 23: I suggest that you show evidence for how you identified these nitrated phenol species. For example, you could show chromatograms of the standards compared with the filter measurements. Otherwise, the reader has to simply trust your identification, which is not good procedure.

Pg. 8 Ln 16-20: It is not clear how you used the PMF receptor model and how you arrived at the solution shown in Fig. 3. Please give more details about the model, including citations for model development. Did you investigate solutions with more/fewer factors? How well does this model capture the trends given the fact that you have only two data points per day?

Sect. 3.2.1: Typically, PMF factors are identified by, e.g., showing a correlation between factor loading and some external tracer. You seem to have assigned factor identifications based on mostly assumptions, rather than by showing evidence. Can you provide more evidence for the identifications? Particularly, can you provide more evidence for the identification of the coal combustion factor, since this was presented

as a 'surprising' result? Otherwise, perhaps you could modify the language to reflect that the identifications that you've given are hypotheses and have some uncertainty.

Pg. 11 Ln. 23: Could it be the case that nitrated phenols and NO2 are simply emitted by the same sources, rather than higher NO2 causing higher nitrated phenol concentrations? I have the same question for the comparisons of NO2 with NSAs and NPs later in this section.

Technical details:

Pg. 3 Ln. 4: Please change "secondary formations" to "secondary formation", here and throughout the manuscript.

Pg. 3 Ln. 14-15: Remove the text "from time to time".

Pg. 4 Ln. 14: Please specify what "TEC" stands for, here and elsewhere.

Pg. 4 Ln. 22: From this line until the end of Sect. 2.1, you should change from present to past tense in order to be consistent with the rest of the text. E.g., change "is" to "was" in this line.

Pg. 5 Ln. 10: "Less frequently" than what? Do you mean "infrequently"?

Pg. 5 Ln. 22: Instead of "restored under", I think you mean "stored at".

Pg. 6 Ln. 20: Change "kinds" to "species", here and elsewhere.

---

## Author Comment (AC1) · 22 Feb 2018

Responses (text in blue) to comments by reviewers (text in black)

Reviewer #1:

This is an interesting study about measurements of nitrated phenols in northern China. Nitrophenols have been analysed and a source apportionment by PMF has been performed identifiying five main contributing source factors. The paper is largely correctly written and contains a wealth of valuable information. Nonetheless, the English language of the manuscript should be checked once more, preferably by a native speaker.

Unfortunately, the analytical finding for the NPs are not being linked to aerosol optical properties which is a topic currently much discussed but maybe this is foreseen for a follow-up or sister publication.

Overall, I think the paper can be published subject to only minor revision.

Reply: Thanks for the recommendation. The manuscript has been obviously improved after revision by native English speakers. The changes are marked in blue in the revised manuscript.

Comments in detail

Page 5. line 6 :....converter...

Reply: Line 12 on Page 4 and Line 10 on Page 5, change "convertor" to "converter".

Page 8, line 14 ff: Please discuss if there are other PMF solutions which do explain equal fractions but derive another number of factors. For Jinan and Wangdu the sum of the five factors is not 100 %. Please mention and discuss.

Reply: The PMF solutions with four and six factors (see Fig. R1 and Fig. R2) are less reasonable and less optimal than the solution with five factors, so the PMF results with five factors are used in this study. For Ji'nan and Wangdu sites, that the sum of the five factors is not 100 % is due to rounding of the decimals.

Page 8, Line 19-24: Based on the results of the PMF model, we evaluated the solutions with four, five, and six factors. The four-factor solution did not distinguish the factor of coal combustion from those of traffic and biomass burning, which failed to provide reasonable separated sources. The six-factor result, however, exhibited two factors with high levels of both SO2 and O3, which indicated splitting from one factor. Therefore, five major factors were finally identified and are shown in Fig. 3.

Figure captions, Figure 5: Scatter plots (in two words)

Reply: Fig. 5, Scatter plots of particulate nitrated phenols with fine inorganic nitrate at

the rural and mountain sites.

Page10, line 8: Please discuss why coal combustion has not been identified before. This may just be due to the fact that this is not a strong source factor contributor either in Shanghai or in Germany.

Reply: As suggested, some discussion on the coal combustion source has been added.

Page 11, Line 10-14: Though there was evidence that coal combustion served as a primary source of nitrated phenols in atmosphere (Lüttke et al., 1997; Kourtchev et al., 2014), it was not identified as a major contributor in East China and eastern Germany in previous studies (Li et al., 2016; Teich et al., 2017). It is probably due to the fact that coal combustion was not an important source contributor in those regions during the measurement periods.

Page 11, section 3.3.: Maybe this section can be substantiated somewhat by a discussion which chemical mechanisms actually lead to the secondary formation. The section is a bit unspecific. The occurrence of which compounds can be explained by pure gas phase processes and where are product observed where multiphase processes could probably be involved? Is there any correlation with haze occurrence?

Reply: We have added some discussion concerning the possible chemical mechanisms in the revised manuscript. Note that high concentrations of nitrated phenols were usually observed with high loadings of particles (i.e., haze pollution) and exhibited good correlation particularly with fine particulate nitrate.

Page 13, Line 13-18: With consideration of the better correlations of nitrated phenols with inorganic nitrates than with NO2, for the selected nocturnal samples mainly from secondary formation, multiphase reactions of precursors on the surfaces of and/or within particles might be the dominant formation pathways of nitrated phenols. Concurrent measurements of gas- and particle-phase nitrated phenols, aromatic precursors,

oxidants, and aerosol physical and chemical properties are needed to clarify the major formation processes.

Reviewer #2:

The authors measured several species of nitrated phenols in PM2.5 filter samples at several sites in North China. A positive matrix factorization (PMF) receptor model was applied to investigate the sources of nitrated phenols, which were found to be traffic, coal combustion, biomass burning, secondary formation, and aged coal combustion. Discussion of the secondary sources of various nitrated phenols was included. I find that this manuscript includes a nice analysis of diurnal, seasonal, and spatial differences in the measured compounds, which are important constituents of organic aerosol. The strength of the manuscript could be greatly improved by including more detail about nitrated phenol species identification, as well as more detail about how the PMF model was run and how the solutions were chosen. I would recommend this paper for publication after considering the following comments.

Reply: Thanks for the Reviewer's constructive and helpful comments. These comments have been addressed in the revised manuscript.

1) We have added some discussion concerning the identification of nitrated phenol species in Section 2.3 in the revised manuscript and Figure S1 in the supplement.

Page 6, Line 17-24: The mass signals at six mass-to-charge ratios (138, 152, 154, 168, 182, and 228 amu) were monitored under the selective ion mode, and the standards of the target compounds and isomers were applied for identification. As shown in Fig. S1, nine species of nitrated phenols were identified: 4-nitrophenol (4NP), 2-methyl-4-nitrophenol (2M4NP), 3-methyl-4-nitrophenol (3M4NP), 4-nitrocatechol (4NC), 4-methyl-5-nitrocatechol (4M5NC), 3-methyl-5-nitrocatechol (3M5NC), 3-methyl-6-nitrocatechol (3M6NC), 3-nitro-salicylic acid (3NSA), and 5-nitro-salicylic acid (5NSA). With the analysis of gradient concentrations of the standard mixtures, standard curves were applied for quantification of the nine nitrated phenols.

2) We have added some discussion concerning the PMF results in Section 3.2.1 in the revised manuscript. The PMF solutions with four and six factors (see Fig. R1 and Fig. R2) are less reasonable and less optimal than the solution with five factors, so the PMF results with five factors are used in this study.

Page 8, Line 19-24: Based on the results of the PMF model, we evaluated the solutions with four, five, and six factors. The four-factor solution did not distinguish the factor of coal combustion from those of traffic and biomass burning, which failed to provide reasonable separated sources. The six-factor result, however, exhibited two factors with high levels of both SO2 and O3, which indicated splitting from one factor. Therefore, five major factors were finally identified and are shown in Fig. 3.

General comments

Title: For clarity, you may want to specify in the title that you measured "particulate nitrated phenols" instead of just "nitrated phenols", as you have done in the abstract and other places.

Reply: Title: Observations of fine particulate nitrated phenols in four sites in North China: Concentrations, source apportionment, and secondary formation.

Pg. 3 Ln. 3: Please add citations to support this sentence.

Reply: A reference has been added.

Harrison, M. A. J., Barra, S., Borghesi, D., Vione, D., Arsene, C., and Iulian Olariu, R.: Nitrated phenols in the atmosphere: a review, Atmos. Environ., 39, 231-248, 10.1016/j.atmosenv.2004.09.044, 2005a.

Pg. 3 Ln. 27-Pg. 4 Ln. 2: These last two sentences in the introduction are actually statements of your results, which should not be included in the introduction. Please consider revising these sentences so that they simply state what you did and are about to present, and not what you found.

[Figure]

Reply: Page 3, Line 24-25: PMF model and correlation analysis are then applied to understand the primary sources and secondary formation of the particulate nitrated phenols in this region.

Pg. 6 Ln 1-2: Why did you multiply by 1.8 and 2.0?

Reply: Page 5, Line 22-25: According to the OM/OC ratios reported in previous studies, organic matter concentrations were calculated from the organic carbon concentrations by multiplying by a factor of 1.8 at the urban Ji'nan site and by a factor of 2.0 at the remote sites of Yucheng and Mt. Tai as an estimation (Aiken et al., 2008; Yao et al., 2016).

Pg. 6 Ln. 23: I suggest that you show evidence for how you identified these nitrated phenol species. For example, you could show chromatograms of the standards compared with the filter measurements. Otherwise, the reader has to simply trust your identification, which is not good procedure.

Reply: Page 6, Line 17-24: The mass signals at six mass-to-charge ratios (138, 152, 154, 168, 182, and 228 amu) were monitored under the selective ion mode, and the standards of the compounds and isomers were applied for identification. As shown in Fig. S1, nine species of nitrated phenols were identified: 4-nitrophenol (4NP), 2-methyl-4-nitrophenol (2M4NP), 3-methyl-4-nitrophenol (3M4NP), 4-nitrocatechol (4NC), 4-methyl-5-nitrocatechol (4M5NC), 3-methyl-5-nitrocatechol (3M5NC), 3-methyl-6-nitrocatechol (3M6NC), 3-nitro-salicylic acid (3NSA), and 5-nitro-salicylic acid (5NSA). With the analysis of gradient concentrations of the standard mixtures, standard curves were applied for quantification of the nine nitrated phenols.

Pg. 8 Ln 16-20: It is not clear how you used the PMF receptor model and how you arrived at the solution shown in Fig. 3. Please give more details about the model, including citations for model development. Did you investigate solutions with more/fewer factors? How well does this model capture the trends given the fact that you have only two data points per day?

Reply: We have added a reference (Paatero and Tapper, 1994) for the PMF model deployed in this study. The PMF solutions with four and six factors (see Fig. R1 and Fig. R2) are less reasonable and less optimal than the solution with five factors, so the PMF results with five factors are used in this study. Total ninety-one sets of input data at four different sites in two seasons were used for the PMF model, so in our view the model can substantially capture the variations of the selected air pollutants.

Page 8, Line 18-24: Ninety-one sets of input data were used and the model was run 40 times to choose the optimal solution. Based on the results of the PMF model, we evaluated the solutions with four, five, and six factors. The four-factor solution did not distinguish the factor of coal combustion from those of traffic and biomass burning, which failed to provide reasonable separated sources. The six-factor result, however, exhibited two factors with high levels of both $SO_2$ and $O_3$, which indicating splitting from one factor. Therefore, five major factors were finally identified and are shown in Fig. 3.

Sect. 3.2.1: Typically, PMF factors are identified by, e.g., showing a correlation be-tween factor loading and some external tracer. You seem to have assigned factor identifications based on mostly assumptions, rather than by showing evidence. Can you provide more evidence for the identifications? Particularly, can you provide more evidence for the identification of the coal combustion factor, since this was presented as a 'surprising' result? Otherwise, perhaps you could modify the language to reflect that the identifications that you've given are hypotheses and have some uncertainty.

Reply: As suggested, more evidences have been provided for the identifications of the major sources of the nitrated phenols observed in North China.

Page 9, Line 5-13: Despite a lack of verification on the direct emission of nitrated phenols from coal combustion at this time, previous field studies provided evidence that coal combustion activities could be an important contributor to the observed elevated levels of particulate nitrated phenols. In field measurements at the summit of Great

Dun Fell, UK, coal combustion was considered to be associated with the relatively high levels of nitrated phenols in cloud (Lüttke et al., 1997). In addition, the large proportion of nitroaromatic compounds in PM2.5 observed in urban Cork, Ireland was attributed to intensive anthropogenic activities including domestic solid fuel burning (peat, coal, and wood) and vehicle emissions (Kourtchev et al., 2014).

Page 9, Line 17-19: The direct emission of nitrated phenols from biomass burning was confirmed and determined by several previous studies, with emission factors ranging from 0.4–11.1 mg kg-1 (Hoffmann et al., 2007; Iinuma et al., 2007; Wang et al., 2017).

Page 9, Line 23-25: Secondary formation was shown to be an important source of nitrated phenols in atmosphere in recent field and modeling studies (Harrison et al., 2005b; Iinuma et al., 2010; Yuan et al., 2016).

Page 10, Line 1-2: The contribution of aged coal combustion plume to the particulate nitrated phenols requires further confirmation and evaluation via chamber simulation and field measurements.

Pg. 11 Ln. 23: Could it be the case that nitrated phenols and NO2 are simply emitted by the same sources, rather than higher NO2 causing higher nitrated phenol concentrations? I have the same question for the comparisons of NO2 with NSAs and NPs later in this section.

Reply: In Section 3.3, only data of samples largely influenced by secondary formation were included and analyzed, by discarding the samples with high contributions (>40%) from primary emission sources (including biomass burning, traffic, and coal combustion) based on the results of PMF model. The relatively low levels of NO2 (see Fig. 7 and Fig. 8) and low concentrations of SO2 (not shown here) also indicate rare influence from primary emission sources to nitrated phenols in the selected samples.

Page 12, Line 14-18: In general, higher concentrations of ΣNPs correlated with higher mixing ratio of NO2 at all four sites in the summertime, and better correlations were

found at the three remote sites than in urban Ji'nan. The relevance of nitrated phenols to NO2 in the rural and mountain areas suggests that NO2 played an important role in the secondary formation of nitrated phenols in North China.

Technical details:

Pg. 3 Ln. 4: Please change "secondary formations" to "secondary formation", here and throughout the manuscript.

Reply: We have changed "secondary formations" to "secondary formation" in the revised manuscript.

Pg. 3 Ln. 14-15: Remove the text "from time to time".

Reply: Page 3, Line 12-13: Once formed in the gas phase, the phenols and nitrated phenols partition between the gas and particle phases according to their saturated vapor pressure.

Pg. 4 Ln. 14: Please specify what "TEC" stands for, here and elsewhere.

Reply: Page 4, Line 11-12: The NOx concentration was measured by a chemiluminescence method equipped with a molybdenum oxide converter (Model 42C, Thermo Electronic Corporation (TEC), USA).

Pg. 4 Ln. 22: From this line until the end of Sect. 2.1, you should change from present to past tense in order to be consistent with the rest of the text. E.g., change "is" to "was" in this line.

Reply: We have checked the grammar in Section 2.1 and corrected the language to past tense.

Pg. 5 Ln. 10: "Less frequently" than what? Do you mean "infrequently"?

Reply: Page 5, Line 6-7: It was infrequently influenced by incense burning and restaurants from the famous tourism spots at Mt. Tai.

Pg. 5 Ln. 22: Instead of "restored under", I think you mean "stored at".

Reply: Page 5, Line 17: The filter samples were stored at ?20°C until subsequent mass weighing and chemical analysis of nitrated phenols, organic carbons, and water-soluble ions.

Pg. 6 Ln. 20: Change "kinds" to "species", here and elsewhere.

Reply: We have changed "kinds" to "species" in the revised manuscript.

Figures:

Figure R1. Source profiles of nitrated phenols and related air pollutants obtained from PMF solution with four factors.

Figure R2. Source profiles of nitrated phenols and related air pollutants obtained from PMF solution with six factors.

Please also note the supplement to this comment:
https://www.atmos-chem-phys-discuss.net/acp-2017-952/acp-2017-952-AC1-supplement.pdf

————————————————

[Figure]

**Supplement:**

**Responses (text in blue) to comments by reviewers (text in black)**

Reviewer #1:

This is an interesting study about measurements of nitrated phenols in northern China. Nitrophenols have been analysed and a source apportionment by PMF has been performed identifiying five main contributing source factors. The paper is largely correctly written and contains a wealth of valuable information. Nonetheless, the English language of the manuscript should be checked once more, preferably by a native speaker. Unfortunately, the analytical finding for the NPs are not being linked to aerosol optical properties which is a topic currently much discussed but maybe this is foreseen for a follow-up or sister publication.

Overall, I think the paper can be published subject to only minor revision.

Reply: Thanks for the recommendation. The manuscript has been obviously improved after revision by native English speakers. The changes are marked in blue in the revised manuscript.

Comments in detail

Page 5. line 6 :....converter...

Reply: Line 12 on Page 4 and Line 10 on Page 5, change "convertor" to "converter".

Page 8, line 14 ff: Please discuss if there are other PMF solutions which do explain equal fractions but derive another number of factors. For Jinan and Wangdu the sum of the five factors is not 100 %. Please mention and discuss.

Reply: The PMF solutions with four and six factors (see Fig. R1 and Fig. R2) are less reasonable and less optimal than the solution with five factors, so the PMF results

with five factors are used in this study. For Ji'nan and Wangdu sites, that the sum of the five factors is not 100 % is due to rounding of the decimals.

Page 8, Line 19-24: Based on the results of the PMF model, we evaluated the solutions with four, five, and six factors. The four-factor solution did not distinguish the factor of coal combustion from those of traffic and biomass burning, which failed to provide reasonable separated sources. The six-factor result, however, exhibited two factors with high levels of both $SO_2$ and $O_3$, which indicated splitting from one factor. Therefore, five major factors were finally identified and are shown in Fig. 3.

Figure captions, Figure 5: Scatter plots (in two words)

Reply: Fig. 5, Scatter plots of particulate nitrated phenols with fine inorganic nitrate at the rural and mountain sites.

Page10, line 8: Please discuss why coal combustion has not been identified before. This may just be due to the fact that this is not a strong source factor contributor either in Shanghai or in Germany.

Reply: As suggested, some discussion on the coal combustion source has been added.

Page 11, Line 10-14: Though there was evidence that coal combustion served as a primary source of nitrated phenols in atmosphere (Lüttke et al., 1997; Kourtchev et al., 2014), it was not identified as a major contributor in East China and eastern Germany in previous studies (Li et al., 2016; Teich et al., 2017). It is probably due to the fact that coal combustion was not an important source contributor in those regions during the measurement periods.

Page 11, section 3.3.: Maybe this section can be substantiated somewhat by a discussion which chemical mechanisms actually lead to the secondary formation. The

section is a bit unspecific. The occurrence of which compounds can be explained by pure gas phase processes and where are product observed where multiphase processes could probably be involved? Is there any correlation with haze occurrence?

Reply: We have added some discussion concerning the possible chemical mechanisms in the revised manuscript. Note that high concentrations of nitrated phenols were usually observed with high loadings of particles (i.e., haze pollution) and exhibited good correlation particularly with fine particulate nitrate.

Page 13, Line 13-18: With consideration of the better correlations of nitrated phenols with inorganic nitrates than with $NO_2$, for the selected nocturnal samples mainly from secondary formation, multiphase reactions of precursors on the surfaces of and/or within particles might be the dominant formation pathways of nitrated phenols. Concurrent measurements of gas- and particle-phase nitrated phenols, aromatic precursors, oxidants, and aerosol physical and chemical properties are needed to clarify the major formation processes.

Reviewer #2:

The authors measured several species of nitrated phenols in PM2.5 filter samples at several sites in North China. A positive matrix factorization (PMF) receptor model was applied to investigate the sources of nitrated phenols, which were found to be traffic, coal combustion, biomass burning, secondary formation, and aged coal combustion. Discussion of the secondary sources of various nitrated phenols was included. I find that this manuscript includes a nice analysis of diurnal, seasonal, and spatial differences in the measured compounds, which are important constituents of organic aerosol. The strength of the manuscript could be greatly improved by including more detail about nitrated phenol species identification, as well as more detail about how the PMF model was run and how the solutions were chosen. I would recommend this paper for publication after considering the following comments.

Reply: Thanks for the Reviewer's constructive and helpful comments. These comments have been addressed in the revised manuscript.

1) We have added some discussion concerning the identification of nitrated phenol species in Section 2.3 in the revised manuscript and Figure S1 in the supplement.

Page 6, Line 17-24: The mass signals at six mass-to-charge ratios (138, 152, 154, 168, 182, and 228 amu) were monitored under the selective ion mode, and the standards of the target compounds and isomers were applied for identification. As shown in Fig. S1, nine species of nitrated phenols were identified: 4-nitrophenol (4NP), 2-methyl-4-nitrophenol (2M4NP), 3-methyl-4-nitrophenol (3M4NP), 4-nitrocatechol (4NC), 4-methyl-5-nitrocatechol (4M5NC), 3-methyl-5-nitrocatechol (3M5NC), 3-methyl-6-nitrocatechol (3M6NC), 3-nitro-salicylic acid (3NSA), and 5-nitro-salicylic acid (5NSA). With the analysis of gradient concentrations of the standard mixtures, standard curves were applied for quantification of the nine nitrated phenols.

2) We have added some discussion concerning the PMF results in Section 3.2.1 in the revised manuscript. The PMF solutions with four and six factors (see Fig. R1 and Fig. R2) are less reasonable and less optimal than the solution with five factors, so the PMF results with five factors are used in this study.

Page 8, Line 19-24: Based on the results of the PMF model, we evaluated the solutions with four, five, and six factors. The four-factor solution did not distinguish the factor of coal combustion from those of traffic and biomass burning, which failed to provide reasonable separated sources. The six-factor result, however, exhibited two factors with high levels of both $SO_2$ and $O_3$, which indicated splitting from one factor. Therefore, five major factors were finally identified and are shown in Fig. 3.

General comments

Title: For clarity, you may want to specify in the title that you measured "particulate

nitrated phenols" instead of just "nitrated phenols", as you have done in the abstract and other places.

Reply: Title: Observations of fine particulate nitrated phenols in four sites in North China: Concentrations, source apportionment, and secondary formation.

Pg. 3 Ln. 3: Please add citations to support this sentence.

Reply: A reference has been added.

Harrison, M. A. J., Barra, S., Borghesi, D., Vione, D., Arsene, C., and Iulian Olariu, R.: Nitrated phenols in the atmosphere: a review, Atmos. Environ., 39, 231-248, 10.1016/j.atmosenv.2004.09.044, 2005a.

Pg. 3 Ln. 27-Pg. 4 Ln. 2: These last two sentences in the introduction are actually statements of your results, which should not be included in the introduction. Please consider revising these sentences so that they simply state what you did and are about to present, and not what you found.

Reply: Page 3, Line 24-25: PMF model and correlation analysis are then applied to understand the primary sources and secondary formation of the particulate nitrated phenols in this region.

Pg. 6 Ln 1-2: Why did you multiply by 1.8 and 2.0?

Reply: Page 5, Line 22-25: According to the OM/OC ratios reported in previous studies, organic matter concentrations were calculated from the organic carbon concentrations by multiplying by a factor of 1.8 at the urban Ji'nan site and by a factor of 2.0 at the remote sites of Yucheng and Mt. Tai as an estimation (Aiken et al., 2008; Yao et al., 2016).

Pg. 6 Ln. 23: I suggest that you show evidence for how you identified these nitrated phenol species. For example, you could show chromatograms of the standards compared with the filter measurements. Otherwise, the reader has to simply trust your identification, which is not good procedure.

Reply: Page 6, Line 17-24: The mass signals at six mass-to-charge ratios (138, 152, 154, 168, 182, and 228 amu) were monitored under the selective ion mode, and the standards of the compounds and isomers were applied for identification. As shown in Fig. S1, nine species of nitrated phenols were identified: 4-nitrophenol (4NP), 2-methyl-4-nitrophenol (2M4NP), 3-methyl-4-nitrophenol (3M4NP), 4-nitrocatechol (4NC), 4-methyl-5-nitrocatechol (4M5NC), 3-methyl-5-nitrocatechol (3M5NC), 3-methyl-6-nitrocatechol (3M6NC), 3-nitro-salicylic acid (3NSA), and 5-nitro-salicylic acid (5NSA). With the analysis of gradient concentrations of the standard mixtures, standard curves were applied for quantification of the nine nitrated phenols.

Pg. 8 Ln 16-20: It is not clear how you used the PMF receptor model and how you arrived at the solution shown in Fig. 3. Please give more details about the model, including citations for model development. Did you investigate solutions with more/fewer factors? How well does this model capture the trends given the fact that you have only two data points per day?

Reply: We have added a reference (Paatero and Tapper, 1994) for the PMF model deployed in this study. The PMF solutions with four and six factors (see Fig. R1 and Fig. R2) are less reasonable and less optimal than the solution with five factors, so the PMF results with five factors are used in this study. Total ninety-one sets of input data at four different sites in two seasons were used for the PMF model, so in our view the model can substantially capture the variations of the selected air pollutants.

Page 8, Line 18-24: Ninety-one sets of input data were used and the model was run 40 times to choose the optimal solution. Based on the results of the PMF model, we

evaluated the solutions with four, five, and six factors. The four-factor solution did not distinguish the factor of coal combustion from those of traffic and biomass burning, which failed to provide reasonable separated sources. The six-factor result, however, exhibited two factors with high levels of both $SO_2$ and $O_3$, which indicating splitting from one factor. Therefore, five major factors were finally identified and are shown in Fig. 3.

Sect. 3.2.1: Typically, PMF factors are identified by, e.g., showing a correlation between factor loading and some external tracer. You seem to have assigned factor identifications based on mostly assumptions, rather than by showing evidence. Can you provide more evidence for the identifications? Particularly, can you provide more evidence for the identification of the coal combustion factor, since this was presented as a 'surprising' result? Otherwise, perhaps you could modify the language to reflect that the identifications that you've given are hypotheses and have some uncertainty.

Reply: As suggested, more evidences have been provided for the identifications of the major sources of the nitrated phenols observed in North China.

Page 9, Line 5-13: Despite a lack of verification on the direct emission of nitrated phenols from coal combustion at this time, previous field studies provided evidence that coal combustion activities could be an important contributor to the observed elevated levels of particulate nitrated phenols. In field measurements at the summit of Great Dun Fell, UK, coal combustion was considered to be associated with the relatively high levels of nitrated phenols in cloud (Lüttke et al., 1997). In addition, the large proportion of nitroaromatic compounds in $PM_{2.5}$ observed in urban Cork, Ireland was attributed to intensive anthropogenic activities including domestic solid fuel burning (peat, coal, and wood) and vehicle emissions (Kourtchev et al., 2014).

Page 9, Line 17-19: The direct emission of nitrated phenols from biomass burning was confirmed and determined by several previous studies, with emission factors ranging from 0.4–11.1 mg kg$^{-1}$ (Hoffmann et al., 2007; Iinuma et al., 2007; Wang et

al., 2017).

Page 9, Line 23-25: Secondary formation was shown to be an important source of nitrated phenols in atmosphere in recent field and modeling studies (Harrison et al., 2005b; Iinuma et al., 2010; Yuan et al., 2016).

Page 10, Line 1-2: The contribution of aged coal combustion plume to the particulate nitrated phenols requires further confirmation and evaluation via chamber simulation and field measurements.

Pg. 11 Ln. 23: Could it be the case that nitrated phenols and NO2 are simply emitted by the same sources, rather than higher NO2 causing higher nitrated phenol concentrations? I have the same question for the comparisons of NO2 with NSAs and NPs later in this section.

Reply: In Section 3.3, only data of samples largely influenced by secondary formation were included and analyzed, by discarding the samples with high contributions (>40%) from primary emission sources (including biomass burning, traffic, and coal combustion) based on the results of PMF model. The relatively low levels of $NO_2$ (see Fig. 7 and Fig. 8) and low concentrations of $SO_2$ (not shown here) also indicate rare influence from primary emission sources to nitrated phenols in the selected samples.

Page 12, Line 14-18: In general, higher concentrations of $\Sigma NPs$ correlated with higher mixing ratio of $NO_2$ at all four sites in the summertime, and better correlations were found at the three remote sites than in urban Ji'nan. The relevance of nitrated phenols to $NO_2$ in the rural and mountain areas suggests that $NO_2$ played an important role in the secondary formation of nitrated phenols in North China.

Technical details:

Pg. 3 Ln. 4: Please change "secondary formations" to "secondary formation", here and throughout the manuscript.

Reply: We have changed "secondary formations" to "secondary formation" in the revised manuscript.

Pg. 3 Ln. 14-15: Remove the text "from time to time".

Reply: Page 3, Line 12-13: Once formed in the gas phase, the phenols and nitrated phenols partition between the gas and particle phases according to their saturated vapor pressure.

Pg. 4 Ln. 14: Please specify what "TEC" stands for, here and elsewhere.

Reply: Page 4, Line 11-12: The $NO_x$ concentration was measured by a chemiluminescence method equipped with a molybdenum oxide converter (Model 42C, Thermo Electronic Corporation (TEC), USA).

Pg. 4 Ln. 22: From this line until the end of Sect. 2.1, you should change from present to past tense in order to be consistent with the rest of the text. E.g., change "is" to "was" in this line.

Reply: We have checked the grammar in Section 2.1 and corrected the language to past tense.

Pg. 5 Ln. 10: "Less frequently" than what? Do you mean "infrequently"?

Reply: Page 5, Line 6-7: It was infrequently influenced by incense burning and restaurants from the famous tourism spots at Mt. Tai.

Pg. 5 Ln. 22: Instead of "restored under", I think you mean "stored at".

Reply: Page 5, Line 17: The filter samples were stored at −20°C until subsequent

mass weighing and chemical analysis of nitrated phenols, organic carbons, and water-soluble ions.

Pg. 6 Ln. 20: Change "kinds" to "species", here and elsewhere.

Reply: We have changed "kinds" to "species" in the revised manuscript.

**Figures:**

[revised manuscript text omitted]